



# Validation of turbulent heat transfer models against eddy covariance flux measurements over a seasonally ice covered lake

Joonatan Ala-Könni, Kukka-Maaria Kohonen, Matti Leppäranta, and Ivan Mammarella

Institute for Atmospheric and Earth System Research / Physics, Faculty of Science, University of Helsinki, Helsinki, Finland

**Correspondence:** Joonatan Ala-Könni (joonatan.ala-konni@helsinki.fi)

**Abstract.** In this study we analyzed turbulent heat fluxes over a seasonal ice cover on boreal lake located in southern Finland. Eddy covariance (EC) measurements from four ice-on seasons between 2014 and 2019 are compared to three different bulk transfer models: one with a constant transfer coefficient, and two with stability adjusted transfer coefficients: the Lake Heat Flux Analyzer and SEA-ICE. All three models correlate to the EC results well in general, although typically underestimating

the magnitude and the variance of the flux in comparison to the EC observations. Differences between the models are small, with the constant transfer coefficient model performing slightly better than the stability adjusted models. Small difference in temperature and humidity between surface and air results in low correlation between models and EC. During melting periods (surface temperature $T_0 > 0°C$), the model performance for LE decreases when comparing to the freezing periods ($T_0 < 0°C$), while the opposite is true for H. At low wind speed EC shows relatively high fluxes ($\pm 20$ W m$^{-2}$) for H and LE due to non-

local effects that the bulk models are not able to reproduce. Finally, the uncertainty in the estimation of the surface temperature and humidity affects the bulk heat fluxes, especially when the difference between surface and air values are small.

## 1 Introduction

According to latest satellite based estimates, there are approximately 117 million lakes larger than 0.002 km$^2$ globally (Verpoorter et al., 2014). About 95 million of these are either above latitude 60°N or below 56°S. The seasonal lake ice zone

extends on the northern hemisphere from 40°N to 80°N (Leppäranta, 2014), so it can safely be estimated that over 80 % of all lakes on Earth receive a seasonal ice cover. A very defining property of lakes with seasonal ice cover is that they display two starkly different states of their surface during the annual cycle. As the ice cover forms, the lake water is effectively isolated from the atmosphere, and the already low amount of short wave radiation inherent for winter is almost completely attenuated in the snow and ice cover (Leppäranta, 2014). Snow/ice–air boundary replaces the water–air boundary, and radical changes in

albedo, emissivity, surface roughness, energy balance and gas exchange occur. Depending on the lake and the local climate, the snow–air boundary layer can be the dominating mode of exchange between the lake and the atmosphere. Thus, understanding the physics of seasonally ice covered lakes is an important, yet often an overlooked aspect, of understanding the behaviour of lakes (Kirillin et al., 2012).

It is easy to understand why ice covered lakes have been overlooked in the past, as most clearly observable activity on lakes

happens during the open water season, but also the remote nature of many of the seasonally ice covered lakes has made their





research difficult from a practical and technical standpoint (Salonen et al., 2009). Nevertheless, while processes in lakes slow down under the ice cover, they do not stop completely. Circulation is driven by sediment heat accumulated there during the summer, meltwater streaming from the surrounding catchment area and solar radiation, with additional mixing produced via the breaking of internal waves promoted by changes in air pressure and wind, and although primary production is minimal, other

biological processes still continue (Hampton et al., 2017) and affect especially the gas fluxes of the lake (Cortés and MacIntyre, 2020). Regardless of the season, lakes have an effect in the very large scale of global climate as well as in the very local scale. In the large scale, lakes affect the global climate by acting as small net sources of carbon ($CO_2$ and $CH_4$) into the atmosphere and also sequestering organic carbon into their sediments from internal biogeochemical processes and from the surrounding environment (Cole et al., 1994). In the small scale, due to their large heat storage, thermal inertia and evaporation, lakes can

significantly affect local weather patterns and microclimate, like rain and snowfall, temperature and cloudiness (Eerola et al., 2014; Ghanbari et al., 2009; Rouse et al., 2005). As lakes contribute significantly to their local climate, understanding their heat balance more precisely has use in many scales, from local short term weather and ice cover forecasting (Ghanbari et al., 2009) to long term global circulation models, where the effect of lakes has been neglected almost completely (Subin et al., 2012).

The yearly cycle of a lake is driven by external forcing, which follow changes in the patterns of the components of the surface energy and water balance. The surface energy balance constitutes of incoming and reflected solar radiation (also known as short wave radiation), incoming and outgoing terrestrial radiation (also called long wave radiation), turbulent heat fluxes (latent and sensible heat flux) and the precipitation heat flux (Kirillin et al., 2012).

Annual changes in the solar radiation drive the changes in seasons, and during the summer it dominates the energy balance.

In autumn the incoming solar radiation decreases every day, and eventually enough heat will be lost through turbulent heat fluxes and outgoing terrestrial radiation to lower the temperature of the water column to the temperature of maximum density, which for fresh water is +4 °C. Then, the lake mixes completely, while cooling continues. At high latitude, seasonal ice cover begins to form usually in the late autumn during clear and low wind conditions associated with anticyclonal weather patterns, although frazil ice can also form in the turbulent surface layer of the lake as well. During nights in calm, cloud free conditions

there is significant loss of heat from the water through long wave radiation and freezing water can accumulate to the surface without being mixed with the warmer water below forming primary ice under which the more permanent congelation ice can form. Later during the winter snow can accumulate over the ice and freeze into solid, opaque snow ice. It insulates the lake more effectively from the solar radiation than the clear congelation ice. Melting begins when the radiation balance turns positive, and the surface absorbs more radiation. The length of ice season varies significantly depending on the local climate at the lake.

Lakes in Southern Finland spend less than half of the year under an ice cover, but above the latitude 65° N lakes are on average have a longer ice-on than ice-off season (Korhonen, 2006). Although statistics can be drawn, every winter on a lake is unique in regards to the length of the ice cover period, the layering of the ice cover, amount on snow accumulation and precipitation.

Radiative components of the energy balance are relatively simple to measure due to the passive instruments with low power consumption required to measure them, but turbulent heat transfer poses more challenges. During last four decades the eddy

covariance (EC) technique has become a very popular method in many fields of environmental and geophysical sciences. It is



an accurate, proven and well established method for directly measuring vertical fluxes of heat, momentum, gases and particles over a wide variety of surface types and ecosystems (Aubinet et al., 2012). Its strong points are the ability to collect long, continuous time series in many different environmental and meteorological conditions. Although it has been extensively used over terrestrial environments, lately EC has been applied in marine and fresh water environments as well.

As the EC setup is not suited for all applications due to the technical complexity of its installation, simpler methods to compute the turbulent heat fluxes from more basic meteorological observations have been developed, with bulk aerodynamic method and profile method being the most popular. They originated from the need to estimate turbulent heat fluxes in situations where only basic meteorological parameters were available, like with remote buoy based oceanographical measurement stations with very low power available. They are also commonly used in global and regional climate models and as well as

in Numerical Weather Prediction models due to their computational simplicity. For different applications (marine, land etc.), the parametrization of the stability, aerodynamic roughness and other parameters of the model can be adjusted accordingly, for example, some models have a completely theoretical approach, while others are empirical.

Estimating turbulent heat fluxes by the bulk aerodynamic method is simpler than measuring them with EC, as only basic meteorological measurements are required, but it inherently contains some limitations and uncertainty. Stable boundary layer

conditions and surface heterogeneity are especially troublesome, and the assumptions made in the Monin-Obukhov similarity theory do not take into account all meteorological phenomena, like non-local effects produced by the boundary between the ice cover and the surrounding forest, present over lake ice cover (Esters et al., 2021; Barskov et al., 2019). Eddy covariance promises to issue some of these deficiencies of the bulk method. As the source area it measures is significantly larger than that of the radiation sensors required by the bulk method, it has the possibility to overcome the challenges of heterogeneous surface.

While few studies have reported short field campaign measurements of EC turbulent heat fluxes over seasonal ice covered lakes (Franz et al., 2018; Barskov et al., 2019), long term turbulent heat flux measurements have not been reported so far. In this study, we present an unique data set collected over a boreal lake in southern Finland over four ice-on seasons between 2014 and 2019. Previous studies of turbulent heat fluxes over lakes have been performed mostly in the open water season, like northern boreal lake in Finnish Lapland (Lohila et al., 2015), boreal lake in Southern Finland (Nordbo et al., 2011) and the

lake in question in this study, Lake Kuivajärvi (Mammarella et al., 2015). Ice-on lake energy balance has been studied, for example, on Lake Kilpisjärvi in NW Finnish Lapland (Leppäranta et al., 2017) and Lake Pääjärvi in Southern Finland (Wang et al., 2005; Jakkila et al., 2009), but these experiments were done without EC equipment and estimated turbulent heat fluxes by bulk aerodynamic formulae and the profile method. EC over seasonal lake ice cover was performed over a thermokarst lake in Siberia (Franz et al., 2018), but it was also only for one winter. Thus, the data set presented here gives us a unique look into

the dynamics of turbulent heat fluxes over seasonal lake ice cover as well as a possibility to validate the functionality of bulk transfer models in this environment.

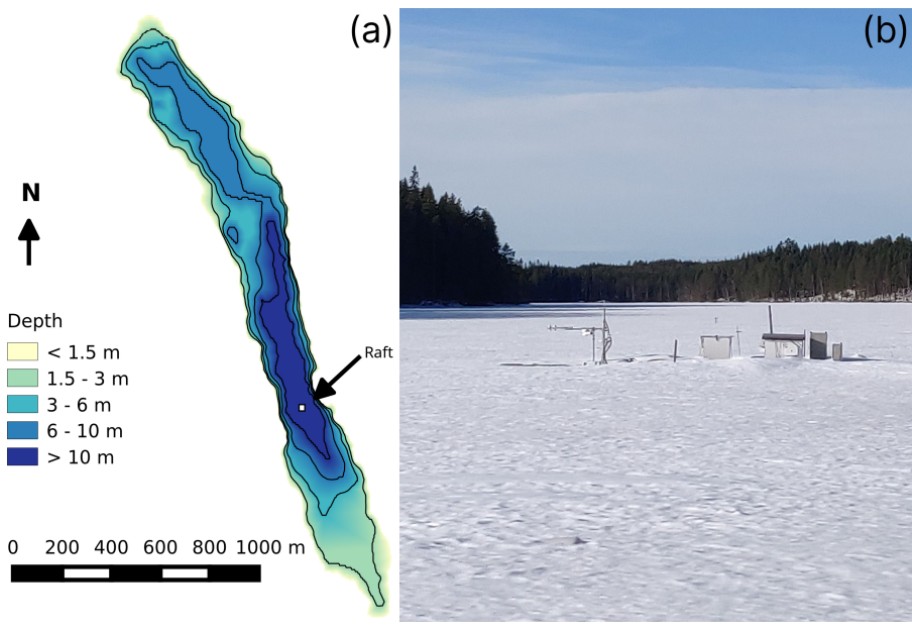

**Figure 1.** Bathymetric map of Lake Kuivajärvi with the position of the EC raft shown as a white square (a). Photo (b) shows the raft as it stood in March 2021. Map adapted from Erkkilä et al. (2018).

## 2 Material & methods

### 2.1 Site

Lake Kuivajärvi is a dimictic, mesotrophic lake in southern Finland (lon. 24° 16' E, lat. 61° 50' N, 141 m above mean sea level). It is located in the Kokemäenjoki water system, which drains into the Baltic Sea. The lake has a strongly elongated shape, 2.6 km in the north – south direction and 200–400 m in the east–west direction (Fig. 1). Due to this shape, wind is usually channeled along the lake. The lake has two basins, the southern basin being the deeper one with a maximum depth of 13.2 m. The raft used for EC and most of the meteorological measurements is located approximately over this deepest point. Lake Kuivajärvi is surrounded mostly by managed Scots pine forest, which is also the home for the SMEAR II station (Hari and Kulmala, 2005). Typical ice cover period in Lake Kuivajärvi lasts for about five months, starting in late November – early December and ending in late April – early May. A mild decreasing trend in the length of the ice-on season has been observed here since the start of observations in 1929 (Korhonen, 2006). The ice cover thickness at the start of the melting period is typically 40–50 cm.





**Table 1.** Ice-on periods at Lake Kuivajärvi used in this study (2014 – 2019) and corresponding number of 30 minute EC flux values used in this study.

| Period | N. of 30 min fluxes (H) & coverage | N. of 30 min fluxes (LE) & coverage |
|---|---|---|
| 9.12.2014 – 20.4.2015 (132 d) | 3 662 (57.8 %) | 3 264 (51.5 %) |
| 23.1.2017 – 3.5.2017 (100 d) | 2 315 (48.2 %) | 1 956 (40.8 %) |
| 3.12.2017 – 23.4.2018 (141 d) | 3 632 (53.7 %) | 2 905 (42.9 %) |
| 16.11.2018 – 25.4.2019 (160 d) | 2 250 (29.3 %) | 1 967 (25.6 %) |
| Total: 533 days | 11 859 (46.3 %) | 10 092 (39.4 %) |

## 2.2 Measurement of fluxes and meteorology

Fluxes of momentum, sensible and latent heat and supporting meteorological measurements were performed on a raft anchored at the deepest point of the lake, as well as on the nearby SMEAR II station (Mammarella et al., 2015). The EC system consists of Metek (Metek GmbH, Elmshorn, Germany) USA-1 3-axis anemometer providing the three component wind speed and sonic temperature and a LI-COR (LI-COR Inc., Nebraska, USA) 7200 measuring the water vapor ($H_2O$) mixing ratio at 10 Hz frequency. A Kipp & Zonen (Kipp & Zonen, Delft, Netherlands) CNR1 net radiometer and pyranometer was used to acquire the radiation balance. This single instrument measured both directions (incoming and outgoing) of the short wave (305 – 2 800 nm) and long wave (5 000 – 50 000 nm) radiation. Relative humidity and air temperature are measured by Rotronic (Rotronic Instrument Corp., NY, USA) MP102H sensor at a height of 1.8 m.

The sonic anemometer and gas inlets were installed at a height of 1.7 m on the western side of the raft, facing sideways of the prevalent wind directions in order to avoid the structure of the raft interfering with the wind and its measurement.

Data from four winters between 2014 and 2019 was used. Ice season was considered to begin on the day when the surface albedo had risen to 0.5, and to end on the day when it had reached values $\alpha < 0.1$ permanently. Dates of ice-on and ice-off for these winters are presented in Table 1.

Surface temperature was derived from outgoing long wave radiation by the Stefan–Boltzmann law, and a constant emissivity of $\epsilon = 0.997$ was assumed, which is a typical value for a snowy surface (Hori et al., 2006).

## 2.3 EC data processing

Eddy covariance raw data were processed with the EddyUH software (Mammarella et al., 2016), and fluxes were calculated using 30 minute averaging time as

$$H = \rho_a c_p \overline{w'T_a'} \tag{1}$$

$$LE = \rho_a L_e \overline{w'q_a'} \tag{2}$$





where $T_a$ is air temperature, $w$ is the vertical wind speed [m s$^{-1}$], $q_a$ is the specific humidity of air [kg kg$^{-1}$], $\rho_a$ is the density of air [kg m$^{-3}$], $L_e$ is the latent heat of evaporation [J kg$^{-1}$] and $c_p$ is the heat capacity of air [J kg$^{-1}$ K$^{-1}$]. The prime marks the fluctuation of the corresponding value from its mean. Micrometeorological notation was used for the sign of the flux: negative fluxes are downward and positive fluxes upwards. State of the art methodologies for the data processing typically

used in land based flux tower (Sabbatini et al., 2018) were applied and adapted following Mammarella et al. (2015). Only evaporation was used instead of evaporation and sublimation in order to be able concentrate more on the errors produced by the calculation of stability.

In short, 2-D coordinate rotation was applied to the anemometer data in order to direct the $u$-component along the mean horizontal wind direction and to result in mean vertical velocity of $\overline{w} = 0$. Linear detrending was applied in the calculation

of turbulent fluctuations. Removal of spikes was performed by setting limits for the difference between subsequent values. Half-hourly blocks of raw data were rejected if they contained over 3 000 spikes. Time lag of H$_2$O was determined from the maximum of cross-covariance function between vertical wind velocity and H$_2$O mixing ratio and cross-wind correction was applied to the sonic temperature data (Liu et al., 2001). High-frequency spectral corrections were done in accordance to (Mammarella et al., 2009). Flux quality flags were based on flux stationarity (FST), skewness (SK) and kurtosis (KU). Only

flux values that had the highest quality flag "0" were used. For this quality class, the conditions were for flux stationarity FST $\leq 0.3$, for skewness -2 < SK < 2 and for kurtosis 1 < KU < 8. Wind direction was also used as a criterion for usable data, and only wind blowing along the lake (130° < WD < 180° and 320° < WD < 350°) were accepted (Erkkilä et al., 2018). A total of 11 859 flux values were accepted for H and 10 092 for LE, resulting in a data coverage of 46.3 % and 39.4 % respectively (Tab. 1).

**2.4 Bulk transfer models**

Various forms of the bulk transfer models have been used to estimate vertical turbulent fluxes for decades. In its simplest form the sensible (latent) heat flux is written as a linear function of wind speed, temperature (humidity) difference between the surface and the air and their corresponding transfer coefficients. In order for the bulk fluxes to have the same sign as the EC data has, in this study they are calculated as

$$H_b = \rho_a c_a C_H (T_0 - T_a) U \quad \text{and} \tag{3}$$

$$LE_b = \rho_a L_e C_E (q_0 - q_a) U \tag{4}$$

where $L_e$ is the heat of vaporization of water (see Eq. 5), $C_H \approx C_E$ are the transfer coefficients of sensible and latent heat,
respectively, $U$ is the wind speed, $T_a$ and $T_0$ are the air and surface temperature respectively and $q_a$ and $q_0$ are the air and surface humidity. In Eq. 4 surface humidity can be assumed to be at saturation due to the watery / icy surface (Leppäranta, 2014).





In this study three different bulk transfer models are compared against EC measurements: one model where the transfer coefficient is kept constant, one developed specifically for open water lake environments (Lake Heat Flux Analyzer) and one developed for sea ice (SHEBA Bulk Turbulent Flux Algorithm for Sea Ice v. 2.0).

### 2.4.1   Constant transfer coefficient model

A constant transfer coefficient model takes in no way into account the stability of atmospheric boundary layer since the values for neutral conditions are applied to all stability conditions. By including this simplified model we can better see the effect of different ways to include the boundary layer stability into the model. Values between $C_E = C_H \approx 1.0 - 1.5 \cdot 10^{-3}$ have been reported over the years for neutral conditions at 10 m height, for example in Kagan (1995). In our study a mean of the values reported in Kagan (1995) was taken and the value was scaled to a height of 1.7 m, resulting in a value $C_E = C_H = 1.8 \cdot 10^{-3}$.

For $L_e$ in the constant coefficient model a temperature dependent model was applied (Rogers, 1989):

$$L_e = 2500.8 - 2.36 T_0 + 0.0016 T_0^2 - 0.00006 T_0^3. \tag{5}$$

### 2.4.2   Lake Heat Flux Analyzer (LHFA)

The second model used in this study is the Lake Heat Flux Analyser software, described in Woolway et al. (2015). It was originally developed in order to create a standardized way to compute turbulent flux values acquired from lake measurement networks. The calculation of roughness length of momentum is based on Smith (1988):

$$z_0 = \left( \frac{\alpha_1 u_{*a}^2}{g} \right) + \left( \frac{\alpha_2 \nu_a}{u_{a*}} \right) \tag{6}$$

where $g$ is gravitational acceleration [m s$^{-2}$], $\alpha_1 = 0.013$ is the Charnock constant, $\alpha_2 = 0.11$, $\nu_a = 1.5 \cdot 10^{-5}$ m$^2$ s$^{-1}$ is kinematic viscosity of air and $u_{*a}$ is the friction velocity in air [m s$^{-1}$]. Roughness lengths [m] of heat ($z_{0T}$) and moisture ($z_{0q}$) are estimated based on Brutsaert (2013) as:

$$z_{0q} = z_{0T} = z_0 \exp(-b_1 Re^{0.25} - b_2), \tag{7}$$

where $b_1 = 2.67$, $b_2 = -2.57$ and $Re = u_{*a} z_0 \nu_a^{-1}$ is roughness Reynolds number. Stability $\zeta$ is calculated as a ratio of measurement height $z$ and Obukhov length $L$, which is defined following (Monin and Obukhov, 1954) as

$$L = \frac{-\rho_a u_{*a}^3 T_v}{kg \left( \frac{H_b}{c_a} + 0.61 \frac{(T_a + 273.16) LE_b}{L_e} \right)} \tag{8}$$



where $T_v = (T_a + 273.16) \cdot (1 + 0.61q_z)$ is the virtual temperature [K], Following stability classes were assigned for the flux-gradient relations for sensible ($\Phi_h(\zeta)$) and latent heat ($\Phi_e(\zeta)$)

$$\Phi_{e,h}(\zeta) = 5 + \zeta \text{ for } \zeta > 1 \text{(very stable)} \tag{9}$$

$$\Phi_{e,h}(\zeta) = 1 + 5\zeta \text{ for } 0 \leq \zeta \leq 1 \text{ (stable)} \tag{10}$$

$$\Phi_{e,h}(\zeta) = (1 - 16\zeta)^{-1/2} \text{ for } -0.465 \leq \zeta \leq 0 \text{ (unstable)} \tag{11}$$

$$\Phi_{e,h}(\zeta) = 0.9k^{4/3}(-\zeta)^{-1/3} \text{ for } \zeta < -0.465 \text{ (very unstable)} \tag{12}$$

where $k = 0.4$ is the von Kármán constant. The final transfer coefficient for latent heat flux for the measurement height $z$ was calculated by

$$C_{Ez} = \frac{-u_{*a}q_*}{u_z(q_0 - q_z)} \tag{13}$$

where $q_*$ is the dimensionless scaling parameter for air humidity, $u_{*a}$ is the friction velocity of air, $u_z$ and $q_z$ are the wind speed and air humidity at height $z$ respectively. The transfer coefficient for sensible heat flux $C_{Hz}$ is calculated similar to this, but replacing the scaling parameter $q_*$ by dimensionless temperature $T_*$ and replacing $(q_0 - q_z)$ by $(T_0 - T_z)$. The software uses an iterative approach to calculate H and LE.

### 2.4.3 SHEBA Bulk Turbulent Flux Algorithm for Sea Ice v. 2.0 (SEA-ICE1)

The third model applied in this study was developed for sea ice and is based on the data set acquired from the SHEBA (Surface Heat Budget of the Arctic Ocean) experiment in the Beaufort Sea (Grachev et al., 2007). In this model, very stable cases ($\zeta > 1$) flux gradient relations rely on empirical approximations from (Grachev et al., 2007), for near-neutral cases no stability correction is applied and for unstable cases formulation from (Paulson, 1970) is applied. The integral forms of stability functions and their corresponding stability classes are as follows:

$$\Psi_{e,h}(\zeta) = \frac{b_h}{2}(1 + c_h\zeta + \zeta^2) + \left(-\frac{a_h}{B_h} + \frac{b_h c_h}{2B_h}\right) \cdot \left(\ln\frac{2\zeta + c_h - B_h}{2\zeta + c_h + B_h} - \ln\frac{c_h - B_h}{c_h + B_h}\right) \text{ for } \zeta > 1 \tag{14}$$





$$\Psi_{e,h}(\zeta) = 0 \text{ for } 0 \leq \zeta \leq 1 \tag{15}$$

$$\Psi_{e,h}(\zeta) = 2\ln\left(\frac{1-x^2}{2}\right) \text{ where } x = (1-\gamma\zeta)^{1/4} \text{ for } \zeta < 0 \tag{16}$$

Here, $a_h = 5$, $b_h = 5$, $c_h = 3$, $B_h = \sqrt{5}$ and $\gamma = 16$. The transfer coefficients for sensible and latent heat fluxes were calculated as

$$C_H = \frac{k^2}{[\ln(z/z_0) - \Psi_m(\zeta)][\ln(z/z_T)] - \Psi_h(z_{0T}/L)} \tag{17}$$

$$C_E = \frac{k^2}{[\ln(z/z_0) - \Psi_m(\zeta)][\ln(z/z_Q)] - \Psi_e(z_{0q}/L)} \tag{18}$$

The roughness lengths were calculated by an analytical method described in Andreas (1987).

The model has settings for both summer and winter sea ice, but for this study it was run in the "winter" setting with the sea ice concentration set at 100 %.

## 3 Results

### 3.1 Environmental drivers of diurnal, seasonal and interannual variation of turbulent heat fluxes

Several meteorological phenomena affect and drive turbulent heat fluxes over a lake ice cover. Gradient in specific humidity or temperature between the surface and the air above it is required for vertical transport of heat or water vapor and wind or convection is needed to keep the air above the surface in movement and generating turbulence. These phenomena do not propagate themselves, but heat is required to drive the wind, bring temperature differences and provide the energy required for evaporation.

Diurnal and seasonal variation as well as the response of the turbulent fluxes to external forcing were studied by dividing the fluxes and meteorological data for all years by month and by hour of day (Fig. 2). Early in the winter (December to February) no diurnal pattern of turbulent fluxes is visible, but as the sun gets higher over the horizon in the spring and short wave radiation begins to dominate the surface energy balance the absolute flux values rise up as well and a diurnal pattern for H (Fig. 2a), LE (Fig. 2b) and wind (Fig. 2h) develops. Thus, the ice-on season can be divided into two phases: early winter with no diurnal pattern and late winter with a diurnal pattern with the sunlight affecting the fluxes.



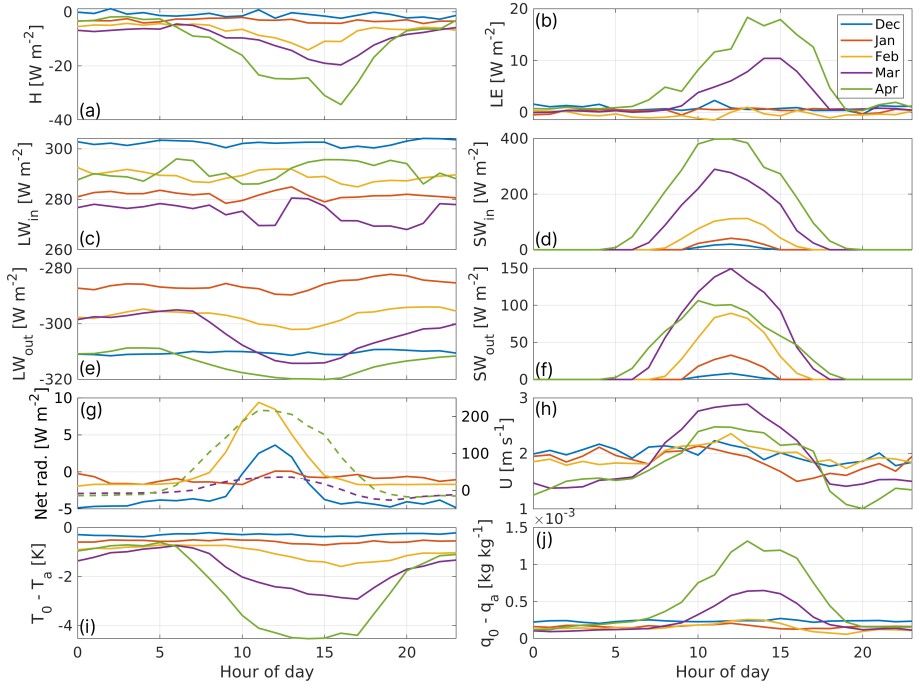

**Figure 2.** Hourly medians of key variables measured over the lake ice: (a) H and (b) LE measured by EC, all four components of the radiation balance separately ((c) through (f)), (g) net radiation, (h) wind speed, (i) temperature difference ($T_0 - T_a$) and (j) specific humidity difference ($q_0 - q_a$). Each curve represents a month of data separated into each hour of the day. Notice, that the turbulent fluxes have the micrometeorological notation, where negative is flux towards the surface, while net radiation has the opposite notation. Net radiation plot has two vertical axes to accommodate the higher fluxes of March and April (dashed lines, scale on the right side), solid lines represent months from December to February (scale on the left). All times are in UTC+2.

The highest median value of sensible heat flux (-34.3 $\mathrm{Wm}^{-2}$,) is seen in April at 16:00 UTC+2 (Fig. 2a). The flux peak is observed at the same time as the peak of temperature gradient (Fig. 2i), and lagging the maximum of net radiation by about four hours (Fig. 2g).

A similar pattern is observable for the latent heat flux (Fig. 2b), but with a positive flux developing along with the intensifying radiation instead of a negative flux. During the darkest winter months there is weak median deposition or sublimation of ice or snow at all hours of the day with no clear diurnal pattern (median $-1.5$ W $\mathrm{m}^{-2} <$ LE $> 1.5$ W $\mathrm{m}^{-2}$, or $\sim \pm 0.05$ mm per day), which turns into daytime evaporation/sublimation in March, when daytime net radiation turns positive. This amount of evaporation or sublimation plays no significant role in the ice and snow mass balance of Lake Kuivajärvi, as the ice cover is in the order of 40 cm with centimeters of snow on top of that, and just the transport of snow by wind can be larger than this evaporation / sublimation. Peak median value, 18.4 W $\mathrm{m}^{-2}$, is in afternoon at 14:00 (UTC+2), following the peak in the humidity difference (Fig. 2j), again lagging some four hours behind the peak in net radiation. Diurnal pattern in the humidity





difference develops later than that of the temperature difference, and hence the diurnal pattern of LE is visible a month later than that of H.

Surface energy balance (Fig. 3f) was calculated as the sum of the four components of radiation and turbulent heat fluxes. Between the months of January and February the surface energy balance had values typically ranging from -50 W m$^{-2}$ and 50 W m$^{-2}$. After March up until ice off surface energy balance varied between -100 W m$^{-2}$ and 700 W m$^{-2}$.

In the darkest winter months the surface temperature of the ice cover follows the air temperature very closely, thus keeping the sensible heat flux low as well. More difference between the two can occur especially in the spring, when the air temperature can reach values well above freezing, but the melting ice and snow surface is stuck at 0 °C, resulting in heat transferred from the air into the melting ice surface. Winters at Lake Kuivajärvi (and in Southern Finland in general) are defined by cold spells separated by days where the air temperature reaches values above freezing. Peaks in sensible heat flux occur then as well, and the number of such peaks varies from year to year.

All timeseries used in this study are presented in Figure 3. Interannual differences in the pattern of turbulent heat fluxes occur mostly in the number of cold and warm periods and differences in their lengths. Low temperatures result in low values of LE for two reasons: one is the very dry nature of cold air and the exponential relation of dew/frost point to air temperature, and the low vertical gradient of specific humidity which follows from this. The second reason is the low wind speeds that are typically associated with the anticyclonal weather patterns that result in low air temperatures during winters. Low air temperatures result in lower values of H as well, but to a smaller degree than for LE. During winters the stability of the atmospheric boundary layer is mostly stable ($zL^{-1} > 0$), as can be seen in Fig. 3e. In this data set stable conditions were measured for 73 % of the time and unstable conditions for 27 %. The change into the higher late-winter flux values (H & LE $> 10$ W m$^{-2}$) quite consistently begins in mid to late March regardless of year. Peaks of $\pm$ 50 W m$^{-2}$ are observed almost daily in spring, while in the early winter they are more intermittent and associated with situations where the air temperature reaches values significantly above freezing.

The onset of ice cover varies from year to year (range of 45 days in the four years studied here), as it is controlled by the temporally random nature of the meteorological conditions suitable for ice formation. Ice-off is more predictable (range of 13 days), as it is driven by the incoming solar radiation which regardless of meteorological factors always begins to dominate the surface energy balance in March.

### 3.2 Comparison of turbulent fluxes derived by EC and bulk transfer models

For the comparison of the EC data with the models the data set was divided into two sets: one where the surface temperature was below freezing and one where it was above freezing. The data are compared in four ways: First, by boxplots where the data are divided, in addition to surface conditions, into each hour of the day (Fig. 4). Second, by scatter plots (Figs. 5 & 7), third by studying the correlation and centered root mean square error (CRMSE) as functions of wind speed, temperature difference and specific humidity difference (Fig. 9), and finally by Taylor plots (Taylor, 2001), showcasing the differences of correlation, CRMSE and standard deviation between the models and the EC measurements (Figs. 6 & 8).



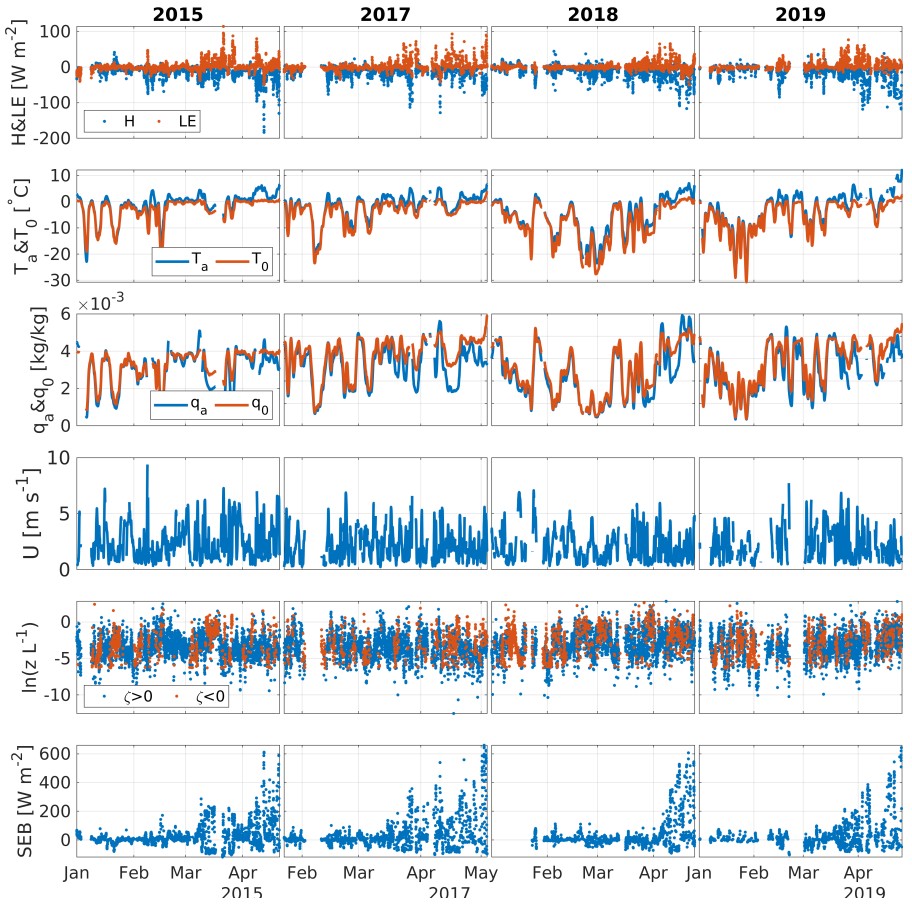

**Figure 3.** All four winters of (a) turbulent flux data, (b) air and surface temperature, (c) air and surface humidity, (d) wind speed, (e) natural logarithm of stability and (f) surface energy balance (SEB). Each column represents one winter. All data are presented as half hourly values, except for wind, which is presented as a six hour moving mean. Notice that the surface energy balance notation is such that positive values indicate heat flux into the surface and vice versa.

### 3.2.1 Sensible heat flux (H)

EC results of sensible heat flux exhibit negative values for most of the time, i.e. heat deposited onto the surface (Figs. 4a & c). During freezing surface conditions (Fig. 4c), the daily cycle of H is not as pronounced as it is during melting conditions, but this is mostly due to the low net radiation values of the early winter months. Frozen conditions result in worse overall agreement between the models and EC and all models are quite consistently underestimating the magnitude of the flux. All models result in same diurnal cycle and differ rather little from each other, but with smaller values and smaller standard deviation than the
EC data.





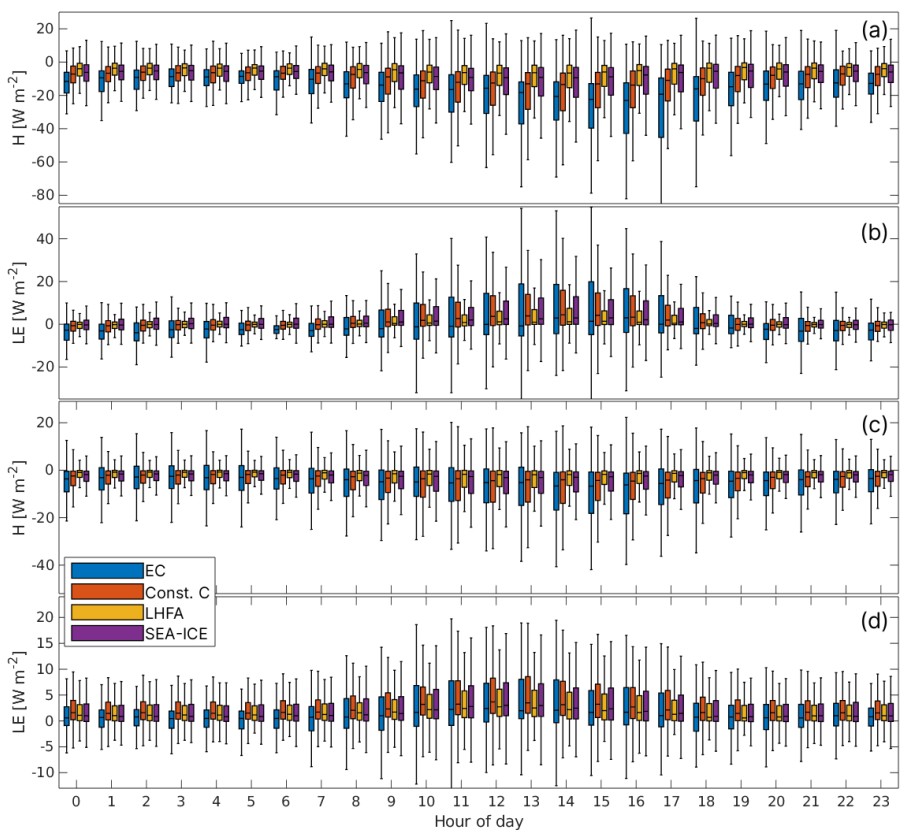

**Figure 4.** Daily variation of fluxes in EC and the three models. (a) H in melting surface conditions, (b) LE in melting surface conditions, (c) H in freezing surface conditions and (d) LE in freezing surface conditions. The line in the middle of each bar indicates median, bar edges indicate 25th and 75th percentiles and the whiskers extending from the bars indicate $2.7\sigma$, or 99.7 % of values are within these boundaries. All times are in UTC+2.

The scatter plots (Fig. 5) reveal how all of the models tend to underestimate the EC fluxes of sensible heat by about 15 – 45 % (Table 2). These plots also reveal the tendency of stability adjusted models to result in near-zero flux, while the EC system measures a significantly non-zero flux. Much higher variability in the model agreement is present in the negative flux values than in positive values, which indicates better agreement during unstable boundary layer conditions. Of the three models,
SEA-ICE performs slightly better than others during unstable conditions (positive flux values for H), as other models show underestimation in comparison to the EC measurements (Fig. 5). Incorrect flux sign reproduction is much more prevalent in freezing surface conditions (Figs. 5a, c & e) than in melting surface conditions (Figs. 5b, d & f).

Taylor plots (Fig. 6) of H reveal, that the simpler, constant $C_H$ model has higher correlation and smaller centered root mean square error (CRMSE) than either of the stability adjusted models when compared with the EC measurements. This is true for
all surface conditions, but melting surface results are in better correlation between models and EC data than melting surface conditions.



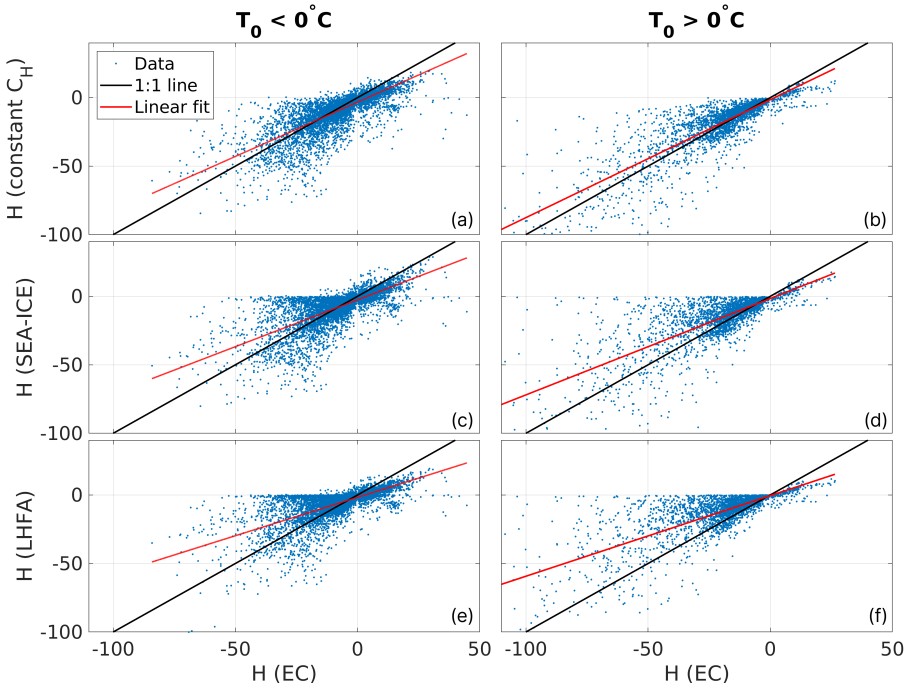

**Figure 5.** Scatter plots of the 30 minute sensible heat flux values of the three included models against corresponding EC measurements. All axes have unit of W m$^{-2}$. Black line indicates a 1:1 fit and red line best linear fit. Linear fit parameters are listed in Table 2.

**Table 2.** Linear fit parameters ($ax + b$) and statistics of 30 minute sensible heat flux (H) values between EC and model data.

|  | Slope $a$ | Offset $b$ [W m$^{-2}$] | Standard deviation [W m$^{-2}$] | CRMSE [W m$^{-2}$] | Correlation |
|---|---|---|---|---|---|
| EC H, $T_0 < 0°$C | - | - | 12.09 | - | - |
| EC H, $T_0 > 0°$C | - | - | 21.30 | - | - |
| H constant $C_H$, $T_0 < 0°$C | 0.79 | -3.29 | 12.28 | 8.08 | 0.78 |
| H constant $C_H$, $T_0 > 0°$C | 0.86 | -1.52 | 21.27 | 11.19 | 0.86 |
| H LHFA $C_H$, $T_0 < 0°$C | 0.56 | -1.69 | 9.33 | 8.89 | 0.68 |
| H LHFA $C_H$, $T_0 > 0°$C | 0.59 | -0.45 | 15.92 | 14.16 | 0.75 |
| H SEA-ICE, $T_0 < 0°$C | 0.68 | -2.55 | 11.37 | 8.69 | 0.73 |
| H SEA-ICE, $T_0 > 0°$C | 0.70 | -1.62 | 18.93 | 13.18 | 0.79 |

### 3.2.2 Latent heat flux (LE)

All three models performed the best for LE in freezing surface conditions, with the sign, magnitude and variability all in fairly good agreement with the EC data (Fig. 4) and small variability between the models (Table 3). The LE flux values are also the
smallest in these conditions.

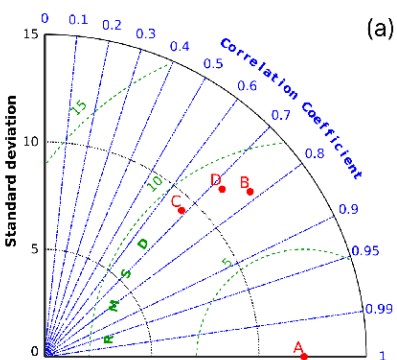
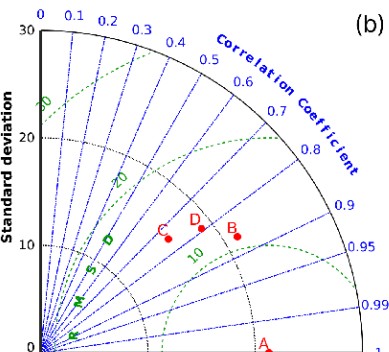

**Figure 6.** Taylor plots comparing H bulk flux models to EC measurements. Subplot (a) has all the cases where the surface was freezing ($T_0 < 0°$C), and in subplot (b) are the cases where the surface was melting ($T_0 > 0°$C). $A$ denotes EC measurements, $B$ is the constant $C_H$ model, $C$ denotes the LHFA model and $D$ stands for the SEA-ICE model. Green lines represent isolines of root mean square error (RMSE), blue lines represent isolines of correlation and dashed black lines are the isolines of standard deviation.

Biggest discrepancies between the models and EC data are found during nights in melting surface conditions (Fig. 4b). Here also the largest differences between the models are present. EC data indicates condensation/deposition on the surface for most of the time during nightly melting surface conditions (median of -3 W m$^{-2}$ and values of up to -15 W m$^{-2}$). These flux values are almost never reproduced by any of the included models, which typically show a median flux of 1 W m$^{-2}$ and peak negative fluxes of -5 W m$^{-2}$. LHFA model performs slightly better than the other two models when LE is negative, but the variability and the magnitude of the fluxes are smaller than the EC measurements. During daytime the range of EC flux values is two to four times greater than the models, with the greatest standard deviation found in the SEA-ICE model and least found in LHFA. The LHFA model has consistently a lower median and smaller variance than the two other models in freezing surface conditions (Figs. 4b & d). The constant $C_E$ model and the SEA-ICE model have very similar performance.

Scatter plots (Fig. 7) reveal that similar pattern of very low bulk fluxes are given when the EC system reports significantly non-zero fluxes. Also, the flux sign is not correctly reproduced sometimes, regardless of surface temperature. The SEA-ICE model results in best linear fit (Table 3), with the static model performing almost as well and LHFA resulting in the most underestimation.

Taylor plots (Fig. 8) of LE show that the three models perform quite similarly in regards to correlation and root mean square error, with greatest differences visible in the standard deviation of the fluxes. By these numbers not one model is clearly better than the others, and they all share the same property of losing some of their predictive ability when the surface is melting. Constant $C_E$ model (Fig. 6b) has a slight advantage in all statistics presented in the Taylor plot over the other two models.





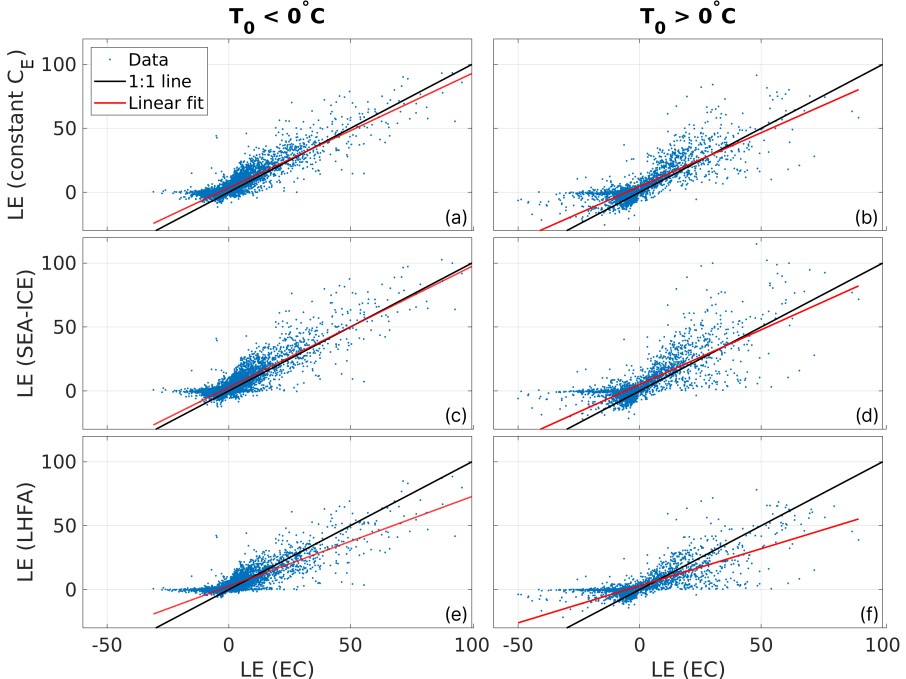

**Figure 7.** Scatter plots of the three included models against latent heat flux EC measurements. Black line indicates a 1:1 fit and red line best linear fit. All axes have unit of W m$^{-2}$. Linear fit parameters are listed in Table 3

.

**Table 3.** Linear fit parameters ($ax + b$) and statistics of 30 minute latent heat flux (LE) values between EC and model data.

|  | Slope $a$ | Offset $b$ [W m$^{-2}$] | Standard deviation [W m$^{-2}$] | CRMSE [W m$^{-2}$] | Correlation |
|---|---|---|---|---|---|
| EC LE, $T_0 < 0°$C | - | - | 9.95 | - | - |
| EC LE, $T_0 > 0°$C | - | - | 14.99 | - | - |
| LE constant $C_E$, $T_0 < 0°$C | 0.90 | 3.39 | 10.03 | 4.73 | 0.89 |
| LE constant $C_E$, $T_0 > 0°$C | 0.84 | 4.53 | 15.30 | 8.98 | 0.82 |
| LE LHFA, $T_0 < 0°$C | 0.70 | 2.65 | 8.18 | 5.13 | 0.86 |
| LE LHFA, $T_0 > 0°$C | 0.58 | 3.00 | 10.85 | 9.78 | 0.76 |
| LE SEA-ICE, $T_0 < 0°$C | 0.95 | 2.75 | 10.74 | 5.19 | 0.88 |
| LE SEA-ICE, $T_0 > 0°$C | 0.86 | 5.01 | 16.43 | 10.48 | 0.78 |

## 3.3 Correlation analysis

The behaviour of the turbulent heat fluxes was also studied by calculating their correlation and centered root mean square error

(CRMSE) as a function of corresponding meteorological variables. This analysis revealed two conditions, where the models



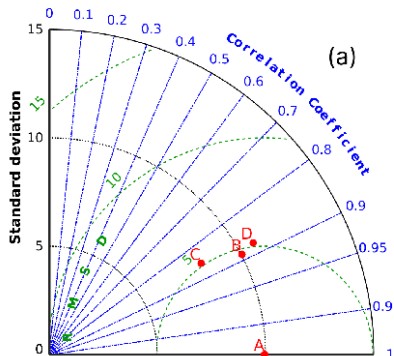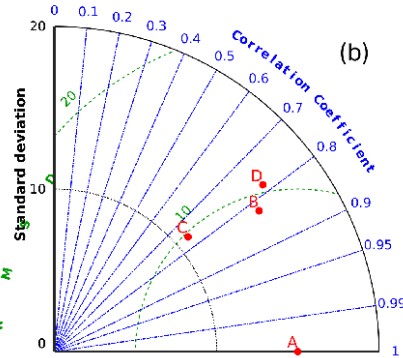

**Figure 8.** Taylor plots comparing LE flux models to EC measurements. Subplot (a) has all the cases where the surface was freezing ($T_0 < 0°C$), and in subplot (b) are the cases where the surface was melting ($T_0 > 0°C$). $A$ denotes EC measurements, $B$ is the constant $C_H$ model, $C$ denotes the LHFA model and $D$ stands for the SEA-ICE model.

have difficulties: low correlation when the temperature/humidity difference between the air and the surface is small and low correlation and high CRMSE during low wind speed.

Figures 9a-c show the correlation and error for sensible heat flux as a function of temperature difference between surface and air ($T_0 - T_a$). A clear depression in correlation can be observed at $\pm 0.5$ K temperature difference for all models. Error for
all models is at its lowest then (CRMSE $< 5$ W m$^{-2}$), as sensible heat flux values themselves are small then as well.

Figures 9d-f shows the correlation and CRMSE of latent heat flux against the difference of specific humidity between surface and air. Similar to the temperature difference, low correlation is observed below 0.2 kg kg$^{-1}$ humidity difference. It can be seen that the correlation remains at around 0 for negative humidity differences as well, i.e. in cases where deposition of water/ice should occur over the lake ice surface. This behaviour was noted previously in Fig. 7 as the inability of the models to reproduce
the scale of nighttime negative EC flux values of LE.

The second type of cases with both high error and low correlation can be found in low wind speed cases (U $\leq 2$ m s$^{-1}$) shown in Figs. 9g and 9h. Bulk models always result in low fluxes in these cases, which follows from the linear dependency of the bulk flux on the wind speed (Eqs. (3) and (4)), but EC measures sometimes relatively high fluxes of around $\pm 20$ W m$^{-2}$ in these conditions. The expected result would have been a similar situation as was with the temperature and humidity:
in low wind conditions correlation drops as the absolute value of the flux drops near zero. In these previously described cases CRMSE would also remain small. Figure 9h shows that correlation drops significantly in low wind conditions, but that CRMSE increases when compared to cases with higher wind speeds (Fig. 9g). This behaviour is present in all stability conditions.

## 4   Discussions & conclusions

Turbulent heat fluxes were studied with an EC setup for four winters over the ice cover of a boreal lake and these results were
compared to three bulk aerodynamic models, one that does not take into account the atmospheric boundary layer stability and



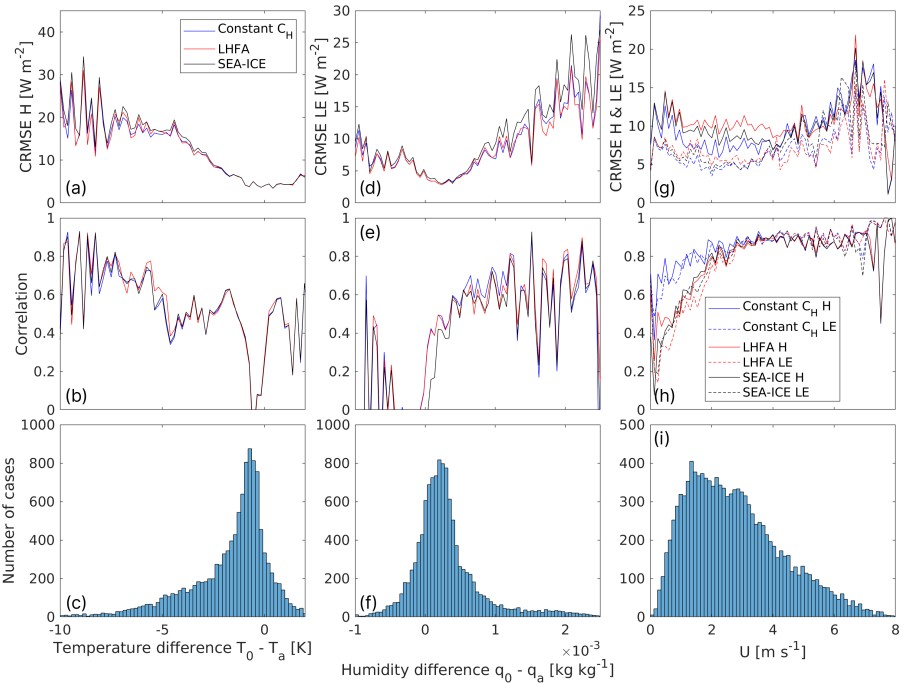

**Figure 9.** The centered root mean square error (CRMSE) (top row) and correlation (middle row) of H (solid line) and LE (dashed line) for 100 equidistant bins of temperature difference, humidity difference and wind speed for each of the three models (red for constant $C_H/E$, blue for LHFA and black for SEA-ICE). Lowest row of figures are histograms of temperature and humidity difference and wind speed values.

two that do take it into account. Since EC measurement sites are much more seldom used than are regular weather stations, bulk transfer models are still going to be used in numerical weather prediction and general circulation models for the foreseeable future. Our data set spanning four ice-on seasons provided a good opportunity to verify and compare the accuracy of bulk transfer models over direct measurements of turbulent heat fluxes by an EC setup.

Both, the bulk transfer models and the EC setup function the best for cases with high wind speeds and large gradients of temperature and humidity, as these conditions give the best agreement between models and EC. Lake ice surface is a challenging environment for eddy covariance due to the relatively low amount of turbulence in the air above it. This is due to several reasons: the boundary layer is stable for most of the time during winter which leads to underdeveloped turbulence and decoupling of the flow from the surface, the surface has a very low roughness and fluxes are usually low. Despite these

problems the EC setup was able to record good quality flux values in a wide range meteorological states, and the data coverage was sufficient. An unfortunate property of the bulk flux calculation was the fact that it functioned the best when the atmospheric boundary layer was unstable, which is the minority of cases studied here.

The bulk aerodynamic method is technically and computationally much simpler than the eddy covariance method, but it comes with some limitations. The greatest error producing effect can be attributed to the fact that estimating the skin tempera-

ture of a snowy surface is difficult, which has been previously reported as a major issue in modeling turbulent heat fluxes over





snow and ice (Franz et al., 2018; Bourassa et al., 2013). This is due to melting and refreezing and the consequent horizontally heterogeneous changes in the surface properties, like albedo, emissivity and phase (liquid or frozen). Errors in determination of the surface temperature affect both sensible and latent heat flux computation especially when the difference between air and surface is small. It results in incorrect surface humidity values, which is an exponential function of the surface temperature
and thus very sensitive to errors. Also, the emissivity $\epsilon$ of the ice/snow surface is difficult to determine accurately, and it can change with the metamorphosis of snow (Hori et al., 2006), which happens constantly over the course of the winter. Thus, the calibration and proper installation of especially the long wave radiation sensors is very important in campaigns performed over ice or snow. Due to the horizontally and vertically heterogeneous nature of the surface, point measurements performed in one location, like net radiation, are not always representative of the whole lake and there is a possibility that very biased results for
the surface albedo and outgoing long wave radiation are recorded, especially during the melting period. This partially explains differences in EC and bulk flux results, as the footprint of EC measurements is at least an order of magnitude larger than the source area measured by the radiation sensors.

Second issue noticed with the bulk algorithms were the low correlation and high error in low wind conditions, which can be explained by non-local effects on turbulence above the lake. In Barskov et al. (2019) it was shown that a sharp decrease
in aerodynamic roughness, like the transition between dense forest and lake ice commonly found on boreal lakes, can cause significant fluxes on EC measurements while bulk algorithms show very low values. When the wind blows from the forest towards the lake, a significant increase in turbulent kinetic energy is observed near the center of the lake, and heat and moisture are transported from the upper boundary layer towards the surface. Bulk algorithms are not taking this phenomenon into account, and thus fail to reproduce these situations. On Lake Kuivajärvi this non-local effect on turbulent fluxes can be observed
at all wind directions due to the small size and strongly elongated shape of the lake and the steep shoreline.

Similar behaviour of the models has been observed in previous studies. In Franz et al. (2018) it was noted that LHFA and SEA-ICE models tended to underestimate and result in lower standard deviation than the turbulent heat fluxes acquired by EC over a Siberian thermokarst lake, which is in line with the results of this study. Correlation of these models ranged between 0.7 and 0.9 over the thermokarst lake, which are similar to our findings.

In a study conducted over landfast sea ice (Raddatz et al., 2015), the bulk transfer models were noticed to underestimate negative fluxes, but unlike our results, were found to overestimate positive fluxes for both H and LE. Correlation coefficients were found to be slightly larger (0.88) for LE than for H (0.82), which is similar to our results. What was also reported in this article is the better accuracy of models with a constant transfer coefficient over dynamic coefficients in the winter - spring transition period, although in general the dynamic model was observed to perform marginally better. In our study the slight
advantage of the static model was present for both fluxes and during winter and spring. Differentiation between frozen and melting surface was not performed in the aforementioned studies, but the results of this study indicate that modeling of H works better in melting conditions, while the opposite holds for LE.

In conclusion, it can be said that in general bulk transfer models are able to reproduce turbulent heat fluxes measured with an EC setup over seasonal lake ice cover for most of the time with relatively little difference between the models tested. In lack
of more accurate measurements of the surface temperature it might not be worth the effort to use stability adjusted models.



While they are more physically sound than the certainly incorrect assumption made in the static model, additional errors are introduced into the flux values from the stability calculations. Turbulent heat flux values obtained by bulk transfer models on very small lakes surrounded by forest should be interpreted with caution, as the heterogeneity of the surrounding environment causes errors in the models.

*Code availability.* The Lake Heat Flux Analyzer (version 1.1.0, released May 19, 2015) is available on Zenodo:
https://doi.org/10.5281/zenodo.5534907
The SEA-ICE (version 2.0, released November 13, 2014) is available on Zenodo:
https://doi.org/10.5281/zenodo.5534911

*Data availability.* All data used in this article can be downloaded from the University of Helsinki Avaa SmartSMEAR database
(https://smear.avaa.csc.fi/).
DOI identifier of the Kuivajärvi data set: doi:10.23729/9b209b52-2ea0-4d89-b059-062b734142d8
DOI identifier for the Hyytiälä forest data set: doi:10.23729/2001890a-2f0b-4e37-8c70-4d2cb5f40273

*Author contributions.* IM designed the study and supervised the research, JAK performed the data analysis and wrote the article, KMK performed the data processing, ML supported data analysis and results interpretation, All authors provide comments to the manuscript.

*Competing interests.* There are no competing interests present in this study.

*Acknowledgements.* The authors thank ICOS-Finland (3119871), ACCC Flagship (337549) and N-PERM project (341348) funded by the Academy of Finland, the Tyumen region government in accordance with the Program of the World-Class West Siberian Inter-regional Scientific and Educational Center (National Project "Nauka") and the European Commission H2020 Research Project RINGO (730944).





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
