# Peer review of "Validation of turbulent heat transfer models against eddy covariance flux measurements over a seasonally ice covered lake"

_Geoscientific Model Development, 2021_

## Author Comment (AC1)

**Responses to referee's comments**

We thank the referees for their valuable comments. Here below we reported our responses to each comment in italics font type.

**Referee 1**

The main drawback of the study is the lack of a detailed analysis and a rather crude approach to using the existing models. At least, a better presentation of the possible sources of errors is needed. This concerns the following issues:

The effect of stability on exchange coefficients and computed fluxes is studied by prescribing constant exchange coefficients and comparing the performance of such a model with other models where the stability-dependent transfer coefficients are used. However, all the models use different roughness lengths, i.e. the neutral transfer coefficients. (The authors should give the exact values of the neutral transfer coefficients of the algorithms, if possible) Thus, one cannot fully separate the effect of stability by comparing their results. Unfortunately, the used observations are limited to one level only and thus it is not possible to fully get rid of the uncertainty associated with the unknown roughness lengths for momentum and heat. The way to proceed would be to prescribe various values of roughness length, or find the best fit based on their data and evaluate the roughness length models.

*As the referee commented, it is not possible to calculate the roughness length accurately from just one level of measurements. The dependency of transfer coefficients on stability parameter z/L (z is the measurement height and L is the Obukhov length) is studied in Fig. 1. Here, the transfer coefficients are normalized for their neutral values (-0.1 < z/L < 0.1) and then binned into 20 bins between stability within range -1 < z/L < 1. The data was filtered for small temperature difference ($T_0$ - $T_a$ < 0.5 K) and very low wind speed (U < 0.5 m/s). Bins were also rejected if they had less than 20 values within them. What can be seen is that the drag coefficients ($C_D$), calculated from measurements and bulk models, follow each other on a decreasing trend in stable conditions (z/L>0) quite well. Instead, the $C_h$ estimated from EC measurements, does not really follow the expected decreasing trend for z/L>0.. This is most likely due to the complex conditions at the site (small lake surrounded by forest) and non-local effects resulting from higher order transport terms for heat, which cause a violation of MOST. Unfortunately, with only one level measurements we are not able to quantify these effects, as was done in Barskov et al (2019) study. In addition, not many cases were recorded during very high stability, as the range of stability values observed over the winters were rather narrow, as is evident from the histograms in Fig. 1. Transfer coefficients for unstable conditions are also difficult to determine due to the predominantly stable conditions prevailing over the winter lake ice cover, which causes noise in the data due to low number of high quality data points.*

The authors use the value 1.8x10$^{-3}$ for the neutral transfer coefficient at 1.7 m height in their most simple bulk model. Is this value representative for a smooth lake ice?

*Over the years many values have been used over smooth ice ranging from 1.0 - 1.5x10$^{-3}$ at 10 m height (Kagan 1995, Elomaa 1977). Value of 1.5x10$^{-3}$ at 10 m height was chosen due to the fact that higher values seemed to perform better in comparison to the EC measurements. This value scales to 1.8x10$^{-3}$ at 1.8 m height.*

What value would result from their own dataset?

*A constant value of $C_H$ could be calculated based on EC measurements, but it was not deemed necessary due to the fact that then the results from this model would not be independent from the EC measurements. The idea was to see how well would a constant $C_H$ model work if a site specific model based on EC could not be constructed. The construction of a model based on lake Kuivajärvi data could be an interesting subject of its own. And the neutral value from the measurements is very close to the one used (see Table 1)*

[Figure]

*Figure 1: Normalized and binned values of transfer coefficients as a function of stability parameter, a) for CD and b) for CH. Subplots c) and d) are corresponding histograms presenting the distribution of transfer coefficient values as a function of stability parameter.*

*Table 1: Neutral values (-0.1 < z/L < 0.1) of the transfer coefficients from the models are as follows (the data will also be added into the manuscript):*

| Model / measurement | Neutral value of bulk coefficients |
|---|---|
| $C_D$ EC | 0.0037 |
| $C_D$ LHFA | 0.0015 |
| $C_D$ SEA-ICE | 0.0019 |
| $C_H$ EC | 0.0019 |
| $C_H$ LHFA | 0.0014 |
| $C_H$ SEA-ICE | 0.0018 |

Why don't the authors try to estimate the roughness length (or the neutral transfer coefficient) for momentum and heat/moisture based on their EC data?

*See the answer above.*

Moreover, roughness length might be dependent on wind direction. The authors do not compare the observed momentum fluxes with those obtained using bulk approach. The question is why? Such an analysis would be helpful in identifying the uncertainties related with roughness length and also stability functions. From their dataset and using best-fit roughness length it is possible to estimate the integral stability correction functions for different z/L (Psi-functions) and compare them

with those prescribed in the bulk models. Such an analysis for the momentum Psi-functions would be free from the uncertainty associated with the uncertainty in the surface temperature. Also, the presented results suggest that increasing roughness length for heat might improve the bulk model performance with respect to sensible heat flux, at least in March-April. The authors should include the sensitivity study to prescribing various roughness lengths.

*As seen from the high value of the drag coefficient ($C_{D\,EC}$) calculated from the measurements (Table 1), the momentum flux may be partially affected by the presence of the raft and the surrounding forest. However, the scalar fluxes are not susceptible to this error, as their calculation does not involve the covariance with the horizontal wind. Related to the stability dependence of $C_D$, it can be seen from Fig. 1 that it follows the expected decreasing trend predicted by MOST.*

The EC fluxes are associated with a certain footprint area. What is the area of such a footprint for the considered site, what types of surface are expected to affect turbulence over the measurement site? Does coastline and forest frequently occur in the footprint area?

*Footprint analysis will be added to the new version of the manuscript. Footprint was calculated in the Matlab script described in Kljun et al (2015). The analysis shows that the 80% limit of the footprint is well within the lake. Limits for the stratification classes were: Stable z/L > 0.1, neutral -0.1 < z/L < 0.1, unstable z/L < -0.1.*

[Figure]

*Figure 2: Footprint of EC measurements calculated from ice-on season 2016/2017 in three stability classes: stable (z/L > 0.1), neutral (-0.1 < z/L < 0.1) and unstable (z/L < -0.1). The plotted lines represent 80 % footprint. Scale is in meters. Calculated by footprint script described in Kljun et al. (2015).*

Why do authors use a bulk model with the roughness length prescribed using Charnock formula for open water? What is their rationale behind that? Of course, for a snow-covered surface a Charnock-like formula for z0 was suggested by Andreas, but can it be applied to the considered lake? The authors do not study this issue.

*Mostly this was just to include different types of models: one optimized for ice, one for open water and one not optimized at all. This is just to show that the choice of the model does not affect the results very much in this case, and they all have error sources in them that are at least in the same scale as the gain given by the optimization. In 6423 cases out of the total of 21953 30 min flux values calculated by the Lake Heat Flux Analyzer the Charnock term was dominating the calculation of $z_0$. This translates to 29 % of the time. As is seen in the statistical comparison of the models (Figs. 6 and 8) the effect of the open water assumption made surprisingly little difference. As the referee has mentioned the use of Charnock-like formula was suggested by Andreas, and we found it is applicable to the considered lake.*

**Referee 2**

1) Equation (14) is not correct. A logarithm is missing in the first term of the r.h.s. Please check your codes if they are correct (with logarithm).

*This was a typo and the code is intact. This will be fixed in the new version of the manuscript.*

2) The main conclusion (see abstract of the manuscript) is that the assumption of a constant transfer coefficient being independent on stratification is the best one. This needs much better explanation. Modellers might get the idea to ignore the stability correction in their runs in general. But this is against all previous experience over decades from observations, theory, and Large Eddy Simulation.

*We agree with the Referee and we will rephrase our main conclusion. Generally, however, we want to say that other sources of uncertainty may be more important. Similar results have been observed in other similar EC setups as well, as I stated in the manuscript.*

So, if this is really the result, then the reader must be better convinced that it is not an artefact. Possible reasons might be conditions violating Monin Obukhov similarity or problems with the accuracy of measurements (e.g. due to influences of the boxes neat the small ‚tower‘?) and many others.

*In conditions such as lake Kuivajärvi it is quite possible that conditions can conflict with Monin Obukhov similarity theory. The forest is quite near the raft, although not in the direct footprint area. We added a reference to the article by Barskov et al. who found that the forest can cause heat fluxes that are in violation of the MO-theory. This seems to be consistent with Fig. 1 showing that the stability dependency of $C_H$ calculated from our data is not the same as predicted from the similarity functions.*

To better convince the reader I find it necessary to show results (fluxes) obtained by the EC method as a function of z/L or of the Richardson number (as in Grachev et al., 2007) or in many other papers (e.g. most recently Srivastava, Gryanik et al.). It would be helpful to show results of the phi- function (eqs. (9,10) and that behind the SHEBA equation (14) (see Grachev et al., 2007) as a function of z/L.

*We have partly shown this in Fig. 1. However, we think that this is not an ideal site (rather very complex) for establishing new stability functions which could be easily used in other studies.*

3) It is difficult to interpret the differences between all three schemes based only on the scatter plots (Figure 5).

*The scatter plots are only one way of showing the differences in the models, and I try to bring out the differences in the other plots as well (Taylor plots, correlation and CRMSE analysis and daily cycles). Full time series is so cluttered that its value in giving information to the reader is somewhat dubious. Although, I am open to new ideas in presenting the data more precisely.*

4) When the final result remains unchanged, it needs to be explained more careful. It should be written that further research is necessary to test the robustness of this result. E.g. measurements over other lakes are necessary before a general suggestion to modellers can be given. Such results could depend on the lake size, where the flow over small lakes might be more inhomogeneous than the flow over large lakes and inhomogeneity might hide the stability dependence.

*This is true, the wording in the current version of the manuscript was drawing too broad conclusions and the referee's comment shall be incorporated into the revised version of the manuscript. The comment about the requirement for new data will also be included in the revised manuscript.*

5) The underestimation of fluxes might be due to errors in the roughness lengths (especially the ratio between roughness length for momentum and for scalars is uncertain).

*The roughness length ratios for the EC data could not be presented, as the measurements were done only at one height. Thus, this is an uncertainty that will remain in this dataset and a mention of this weakness will be added.*

**References**

**Barskov, Kirill & Stepanenko, V. & Repina, Irina & Artamonov, Arseny & Gavrikov, Alexander. (2019). Two Regimes of Turbulent Fluxes Above a Frozen Small Lake Surrounded by Forest. Boundary-Layer Meteorology. 173. 10.1007/s10546-019-00469-w.**

**Elomaa, Esko. *Pääjärvi Representative Basin in Finland: Heat Balance of a Lake*. Hki: [Helsingin yliopisto], 1977.**

**Kagan, B. (1995). *Ocean Atmosphere Interaction and Climate Modeling* (Cambridge Atmospheric and Space Science Series) (M. Hazin, Trans.). Cambridge: Cambridge University Press. doi:10.1017/CBO9780511628931**

**Kljun, N., Calanca, P., Rotach, M. W., and Schmid, H. P.: A simple two-dimensional parameterisation for Flux Footprint Prediction (FFP), Geosci. Model Dev., 8, 3695–3713, https://doi.org/10.5194/gmd-8-3695-2015, 2015.**

---

## Author Comment (AC2)

**Responses to referee's comments**

We thank the referees for their valuable comments. Here below we reported our responses to each comment in italics font type.

**Referee 1**

The main drawback of the study is the lack of a detailed analysis and a rather crude approach to using the existing models. At least, a better presentation of the possible sources of errors is needed. This concerns the following issues:

The effect of stability on exchange coefficients and computed fluxes is studied by prescribing constant exchange coefficients and comparing the performance of such a model with other models where the stability-dependent transfer coefficients are used. However, all the models use different roughness lengths, i.e. the neutral transfer coefficients. (The authors should give the exact values of the neutral transfer coefficients of the algorithms, if possible) Thus, one cannot fully separate the effect of stability by comparing their results. Unfortunately, the used observations are limited to one level only and thus it is not possible to fully get rid of the uncertainty associated with the unknown roughness lengths for momentum and heat. The way to proceed would be to prescribe various values of roughness length, or find the best fit based on their data and evaluate the roughness length models.

*As the referee commented, it is not possible to calculate the roughness length accurately from just one level of measurements. The dependency of transfer coefficients on stability parameter z/L (z is the measurement height and L is the Obukhov length) is studied in Fig. 1. Here, the transfer coefficients are normalized for their neutral values (-0.1 < z/L < 0.1) and then binned into 20 bins between stability within range -1 < z/L < 1. The data was filtered for small temperature difference ($T_0$ - $T_a$ < 0.5 K) and very low wind speed (U < 0.5 m/s). Bins were also rejected if they had less than 20 values within them. What can be seen is that the drag coefficients ($C_D$), calculated from measurements and bulk models, follow each other on a decreasing trend in stable conditions (z/L>0)  quite well. Instead, the $C_h$ estimated from EC measurements, does not really follow the expected decreasing trend for z/L>0.. This is most likely due to the complex conditions at the site (small lake surrounded by forest) and non-local effects resulting from higher order transport terms for heat, which cause a violation of MOST. Unfortunately, with only one level measurements we are not able to quantify these effects, as was done in Barskov et al (2019) study. In addition, not many cases were recorded during very high stability, as the range of stability values observed over the winters were rather narrow, as is evident from the histograms in Fig. 1. Transfer coefficients for unstable conditions are also difficult to determine due to the predominantly stable conditions prevailing over the winter lake ice cover, which causes noise in the data due to low number of high quality data points.*

The authors use the value $1.8 \times 10^{-3}$ for the neutral transfer coefficient at 1.7 m height in their most simple bulk model. Is this value representative for a smooth lake ice?

*Over the years many values have been used over smooth ice ranging from 1.0 - $1.5 \times 10^{-3}$ at 10 m height (Kagan 1995, Elomaa 1977). Value of $1.5 \times 10^{-3}$ at 10 m height was chosen due to the fact that higher values seemed to perform better in comparison to the EC measurements. This value scales to $1.8 \times 10^{-3}$ at 1.8 m height.*

What value would result from their own dataset?

*A constant value of $C_H$ could be calculated based on EC measurements, but it was not deemed necessary due to the fact that then the results from this model would not be independent from the EC measurements. The idea was to see how well would a constant $C_H$ model work if a site specific model based on EC could not be constructed. The construction of a model based on lake Kuivajärvi data could be an interesting subject of its own. And the neutral value from the measurements is very close to the one used (see Table 1)*

[Figure]

*Figure 1: Normalized and binned values of transfer coefficients as a function of stability parameter, a) for CD and b) for CH. Subplots c) and d) are corresponding histograms presenting the distribution of transfer coefficient values as a function of stability parameter.*

Table 1: Neutral values (-0.1 < z/L < 0.1) of the transfer coefficients from the models are as follows (the data will also be added into the manuscript):

| Model / measurement | Neutral value of bulk coefficients |
|---|---|
| $C_D$ EC | 0.0037 |
| $C_D$ LHFA | 0.0015 |
| $C_D$ SEA-ICE | 0.0019 |
| $C_H$ EC | 0.0019 |
| $C_H$ LHFA | 0.0014 |
| $C_H$ SEA-ICE | 0.0018 |

Why don't the authors try to estimate the roughness length (or the neutral transfer coefficient) for momentum and heat/moisture based on their EC data?

*See the answer above.*

Moreover, roughness length might be dependent on wind direction. The authors do not compare the observed momentum fluxes with those obtained using bulk approach. The question is why? Such an analysis would be helpful in identifying the uncertainties related with roughness length and also stability functions. From their dataset and using best-fit roughness length it is possible to estimate the integral stability correction functions for different z/L (Psi-functions) and compare them

with those prescribed in the bulk models. Such an analysis for the momentum Psi-functions would be free from the uncertainty associated with the uncertainty in the surface temperature. Also, the presented results suggest that increasing roughness length for heat might improve the bulk model performance with respect to sensible heat flux, at least in March-April. The authors should include the sensitivity study to prescribing various roughness lengths.

*As seen from the high value of the drag coefficient ($C_{D\,EC}$) calculated from the measurements (Table 1), the momentum flux may be partially affected by the presence of the raft and the surrounding forest. However, the scalar fluxes are not susceptible to this error, as their calculation does not involve the covariance with the horizontal wind. Related to the stability dependence of $C_D$, it can be seen from Fig. 1 that it follows the expected decreasing trend predicted by MOST.*

The EC fluxes are associated with a certain footprint area. What is the area of such a footprint for the considered site, what types of surface are expected to affect turbulence over the measurement site? Does coastline and forest frequently occur in the footprint area?

*Footprint analysis will be added to the new version of the manuscript. Footprint was calculated in the Matlab script described in Kljun et al (2015). The analysis shows that the 80% limit of the footprint is well within the lake. Limits for the stratification classes were: Stable z/L > 0.1, neutral - 0.1 < z/L < 0.1, unstable z/L < -0.1.*

[Figure]

*Figure 2: Footprint of EC measurements calculated from ice-on season 2016/2017 in three stability classes: stable (z/L > 0.1), neutral (-0.1 < z/L < 0.1) and unstable (z/L < -0.1). The plotted lines represent 80 % footprint. Scale is in meters. Calculated by footprint script described in Kljun et al. (2015).*

Why do authors use a bulk model with the roughness length prescribed using Charnock formula for open water? What is their rationale behind that? Of course, for a snow-covered surface a Charnock-like formula for z0 was suggested by Andreas, but can it be applied to the considered lake? The authors do not study this issue.

*Mostly this was just to include different types of models: one optimized for ice, one for open water and one not optimized at all. This is just to show that the choice of the model does not affect the results very much in this case, and they all have error sources in them that are at least in the same scale as the gain given by the optimization. In 6423 cases out of the total of 21953 30 min flux values calculated by the Lake Heat Flux Analyzer the Charnock term was dominating the calculation of $z_0$. This translates to 29 % of the time. As is seen in the statistical comparison of the models (Figs. 6 and 8) the effect of the open water assumption made surprisingly little difference. As the referee has mentioned the use of Charnock-like formula was suggested by Andreas, and we found it is applicable to the considered lake.*

**Referee 2**

1) Equation (14) is not correct. A logarithm is missing in the first term of the r.h.s. Please check your codes if they are correct (with logarithm).

*This was a typo and the code is intact. This will be fixed in the new version of the manuscript.*

2) The main conclusion (see abstract of the manuscript) is that the assumption of a constant transfer coefficient being independent on stratification is the best one. This needs much better explanation. Modellers might get the idea to ignore the stability correction in their runs in general. But this is against all previous experience over decades from observations, theory, and Large Eddy Simulation.

*We agree with the Referee and we will rephrase our main conclusion. Generally, however, we want to say that other sources of uncertainty may be more important. Similar results have been observed in other similar EC setups as well, as I stated in the manuscript.*

So, if this is really the result, then the reader must be better convinced that it is not an artefact. Possible reasons might be conditions violating Monin Obukhov similarity or problems with the accuracy of measurements (e.g. due to influences of the boxes neat the small ‚tower'?) and many others.

*In conditions such as lake Kuivajärvi it is quite possible that conditions can conflict with Monin Obukhov similarity theory. The forest is quite near the raft, although not in the direct footprint area. We added a reference to the article by Barskov et al. who found that the forest can cause heat fluxes that are in violation of the MO-theory. This seems to be consistent with Fig. 1 showing that the stability dependency of $C_H$ calculated from our data is not the same as predicted from the similarity functions.*

To better convince the reader I find it necessary to show results (fluxes) obtained by the EC method as a function of z/L or of the Richardson number (as in Grachev et al., 2007) or in many other papers (e.g. most recently Srivastava, Gryanik et al.). It would be helpful to show results of the phi- function (eqs. (9,10) and that behind the SHEBA equation (14) (see Grachev et al., 2007) as a function of z/L.

*We have partly shown this in Fig. 1. However, we think that this is not an ideal site (rather very complex) for establishing new stability functions which could be easily used in other studies.*

3) It is difficult to interpret the differences between all three schemes based only on the scatter plots (Figure 5).

*The scatter plots are only one way of showing the differences in the models, and I try to bring out the differences in the other plots as well (Taylor plots, correlation and CRMSE analysis and daily cycles). Full time series is so cluttered that its value in giving information to the reader is somewhat dubious. Although, I am open to new ideas in presenting the data more precisely.*

4) When the final result remains unchanged, it needs to be explained more careful. It should be written that further research is necessary to test the robustness of this result. E.g. measurements over other lakes are necessary before a general suggestion to modellers can be given. Such results could depend on the lake size, where the flow over small lakes might be more inhomogeneous than the flow over large lakes and inhomogeneity might hide the stability dependence.

*This is true, the wording in the current version of the manuscript was drawing too broad conclusions and the referee's comment shall be incorporated into the revised version of the manuscript. The comment about the requirement for new data will also be included in the revised manuscript.*

5) The underestimation of fluxes might be due to errors in the roughness lengths (especially the ratio between roughness length for momentum and for scalars is uncertain).

*The roughness length ratios for the EC data could not be presented, as the measurements were done only at one height. Thus, this is an uncertainty that will remain in this dataset and a mention of this weakness will be added.*

Minor comments

Line 73 – all the models are to some extent semi-empirical. Better to avoid such a sentence without explicit explanation what is meant.

*True. This has been corrected.*

Line 119 – what is the sensitivity of the output of bulk algorithms to the used value of surface emissivity varied in the range of natural variability? For example, how much would the fluxes change if eps = 0.98 is used?

*Optimizing the value of emissivity could be used to improve the results. Within the range of e = 0.98 – 1 the difference in the final results is small (approx. 5 – 10 % change in RMSE, standard deviation and correlation) and does not change the final conclusions of the manuscript: the static bulk transfer model still performs marginally better than the dynamic models and the difference between the models remains small. Table 2 presents values of CRMSE and correlation of models with two dfferent values of emissivity ($\varepsilon$ = 0.98 & 0.997). Mention of this has been added to the manuscript.*

*Table 2: Centered root mean square error (CRMSE) and correlation coefficient of the models used in the manuscript for two values of emissivity ($\varepsilon$ = 0.98 & 0.997).*

| Model | CRMSE ($\varepsilon$ = 0.997) [W m$^{-2}$] | CRMSE ($\varepsilon$ = 0.98) [W m$^{-2}$] | Correlation ($\varepsilon$ = 0.997) | Correlation ($\varepsilon$ = 0.98) |
|---|---|---|---|---|
| Static H, freezing | 10.23 | 10.93 | 0.78 | 0.78 |
| Static H, melting | 11.19 | 9.28 | 0.86 | 0.87 |
| LHFA H, freezing | 9.33 | 9.27 | 0.68 | 0.66 |
| LHFA H, melting | 14.16 | 12.32 | 0.75 | 0.76 |
| SEA-ICE H, freezing | 11.37 | 11.30 | 0.73 | 0.70 |
| SEA-ICE H, melting | 13.18 | 11.34 | 0.79 | 0.80 |

| | | | | |
|---|---|---|---|---|
| Static LE, freezing | 4.73 | 5.62 | 0.89 | 0.86 |
| Static LE, melting | 8.98 | 8.61 | 0.82 | 0.83 |
| LHFA LE, freezing | 5.13 | 4.59 | 0.86 | 0.86 |
| LHFA LE, melting | 9.78 | 8.44 | 0.76 | 0.79 |
| SEA-ICE LE, freezing | 5.19 | 4.62 | 0.88 | 0.88 |
| SEA-ICE LE, melting | 10.48 | 8.94 | 0.78 | 0.81 |

Line 200 T* is not a dimensionless temperature, but it is the temperature scale which has dimension [K]

*True. This has been corrected.*

Lines 230-235 it should be better explained how the increase of solar radiation results in negative heat flux and T0-Ta at daytime in spring. Obviously, this is only possible if somewhere around the lake the daytime heat flux and T0-Ta become positive.

*I elaborated the results a little to make it more clear, but in my opinion the results are sensible. Increasing short wave radiation results in positive LE (i.e. ice cover evaporating) and negative H (i.e. heat transferred into the surface) due to the fact that the air temperature is higher that the melting surface which remains near 0 °C for as long as there is any ice left to melt.*

**References**

**Barskov, Kirill & Stepanenko, V. & Repina, Irina & Artamonov, Arseny & Gavrikov, Alexander. (2019). Two Regimes of Turbulent Fluxes Above a Frozen Small Lake Surrounded by Forest. Boundary-Layer Meteorology. 173. 10.1007/s10546-019-00469-w.**

**Elomaa, Esko. *Pääjärvi Representative Basin in Finland: Heat Balance of a Lake*. Hki: [Helsingin yliopisto], 1977.**

**Kagan, B. (1995). *Ocean Atmosphere Interaction and Climate Modeling* (Cambridge Atmospheric and Space Science Series) (M. Hazin, Trans.). Cambridge: Cambridge University Press. doi:10.1017/CBO9780511628931**

**Kljun, N., Calanca, P., Rotach, M. W., and Schmid, H. P.: A simple two-dimensional parameterisation for Flux Footprint Prediction (FFP), Geosci. Model Dev., 8, 3695–3713, https://doi.org/10.5194/gmd-8-3695-2015, 2015.**

---

## Author Response (AR1)

*Author Joonatan Ala-Könni on 19th of May, 2022*

Details of the changes in the manuscript can be seen in the track-changes file, but in summary the following changes were made based on the feedback from referees:

- Addition of new subsection *"Transfer coefficients"* in the section *"Results"* regarding the analysis of the transfer coefficients with two new figures (Figs. 11 & 12) and a table of transfer coefficients (Table 4). Please note, that while Fig. 12 is similar to the figure presented in our answers to the referee's comments it has been improved with better data filtering.

- Addition of new subsection *"Footprint analysis"* in the *"Materials and methods"* section with the accompanying figure 2.

- Conclusions were revised to not contain as strong language as it originally had.

- Model descriptions simplified in *"Material and methods"* section.

- Error in Fig. 3e fixed (subplot upside down).

- Numerous small improvements of clarity, language and spelling throughout the manuscript.